# Diagnostic Predictors of Immunotherapy Response in Head and Neck Squamous Cell Carcinoma

**DOI:** 10.3390/diagnostics13050862

**Published:** 2023-02-23

**Authors:** Piero Giuseppe Meliante, Federica Zoccali, Marco de Vincentiis, Massimo Ralli, Carla Petrella, Marco Fiore, Antonio Minni, Christian Barbato

**Affiliations:** 1Department of Sense Organs, Sapienza University of Rome, 00161 Roma, Italy; 2Institute of Biochemistry and Cell Biology (IBBC), National Research Council (CNR), Department of Sense Organs, Sapienza University of Rome, Viale del Policlinico 155, 00161 Roma, Italy; 3Division of Otolaryngology-Head and Neck Surgery, Ospedale San Camillo de Lellis, ASL Rieti-Sapienza University, Viale Kennedy, 02100 Rieti, Italy

**Keywords:** head and neck squamous cell carcinoma, immunotherapy, PD-1/PD-L1, immunotherapy molecular marker, immunotherapy resistance, pembrolizumab, nivolumab, chemotherapy

## Abstract

Programmed cell death ligand-1 (PD-L1) binds PD-1 on CD^8+^ lymphocytes, inhibiting their cytotoxic action. Its aberrant expression by head and neck squamous cell carcinoma (HNSCC) cells leads to immune escape. Pembrolizumab and nivolumab, two humanized monoclonal antibodies against PD-1, have been approved in HNSCC treatment, but ~60% of patients with recurrent or metastatic HNSCC fail to respond to immunotherapy and only 20 to 30% of treated patients have long-term benefits. The purpose of this review is to analyze all the fragmentary evidence present in the literature to identify what future diagnostic markers could be useful for predicting, together with PD-L1 CPS, the response to immunotherapy and its durability. We searched PubMed, Embase, and the Cochrane Register of Controlled Trials and we summarize the evidence collected in this review. We confirmed that PD-L1 CPS is a predictor of response to immunotherapy, but it should be measured across multiple biopsies and repeatedly over time. PD-L2, IFN-γ, EGFR, VEGF, TGF–β, TMB, blood TMB, CD73, TILs, alternative splicing, tumor microenvironment, and some macroscopic and radiological features are promising predictors worthy of further studies. Studies comparing predictors appear to give greater potency to TMB and CXCR9.

## 1. Introduction

Programmed cell death ligand-1 (PD-L1) is a physiologically expressed transmembrane molecule crucial in immune tolerance. The programmed cell death-1 (PD-1) molecule binds on the surface of CD^8+^ T lymphocytes, blocking their cytotoxic action [1,2,3]. The aberrant expression of PD-L1 by head and neck squamous cell carcinoma (HNSCC) cells inhibits the cytotoxic activity of T cells, leading to immune escape [4,5]. After this discovery, several antibodies active against the PD-1/PD-L1 axis were tested. Large phase III trials (KEYNOTE-040, KEYNOTE-048, CheckMate-141) demonstrated that immune checkpoint inhibitors (ICIs) against PD-1 and PD-L1 outperformed the past gold standard therapy in terms of oncologic outcomes in the treatment of recurrent or metastatic (R/M) HNSCC [4,6,7]. Pembrolizumab (Keytruda^TM^, Merck & Co., Inc., Rahway, NJ, USA) and nivolumab (OPDIVO^TM^, Bristol-Myers Squibb Company, New York, NY, USA), two humanized monoclonal antibodies against PD-1, have been approved in HNSCC treatment by the U.S. Food and Drug Administration (FDA) and the European Medical Agency (EMA) [8]. However, ~60% of patients with recurrent or metastatic HNSCC fail to respond to immunotherapy and only 20 to 30% of treated patients have long-term benefits from ICIs [4,8,9].

Despite the enormous development that immunotherapy is having, the tools for selecting candidates for it have not evolved hand in hand [10,11]. The marker currently used to predict response to therapy is the combined positive score (CPS) for PD-L1 expression by the neoplastic cell (calculated as 100 times the number of PD-L1-positive cancer cells, lymphocytes, and macrophages, divided by the number of viable tumor cells). It is well known from the literature that CPS > 1% is related to immunotherapy response [4,6,7]. Whereby, only patients with CPS > 1 are treated with anti-PD-1/L1 drugs [9]. This predictor does not always work, and above all, it often does not indicate a long-term response. Furthermore, there is still a proportion of patients, as evidenced in large clinical studies, who, despite being below this threshold, would still respond to immunotherapy [4,6,7,8,9,12].

We aim to analyze all the fragmentary evidence presented in the literature to identify what future diagnostic markers could be useful for predicting, together with PD-L1 CPS, the response to immunotherapy and its durability, as well as identifying the most suitable therapy for the individual patient.

## 2. Materials and Methods

We searched the PubMed, Embase, and Cochrane Register of Controlled Trials databases for markers that could predict response to immunotherapy treatment as well as treatment selection. The keywords used by the authors were: head and neck squamocellular carcinoma or head and neck squamous cell carcinoma, immunotherapy, pembrolizumab, nivolumab, markers of response to immunotherapy, predictors of response to immunotherapy, immune checkpoint inhibitors. We have only considered articles in English and without time limits or restrictions on the type of publication. We first screened for titles and abstracts and then read the body of the selected articles. Within the latter, we also manually searched the bibliography for relevant manuscripts. We have decided not to include findings from other types of cancer and treatments not tested on HNSCC. Similarly, we decided not to include discoveries made only in animal models as they are still too far from clinical applicability. In the end, after several meetings between all the authors in which we discussed the results of our research, we summarized the evidence collected within the present manuscript.

## 3. Results

PD-L1 expression is predictive of response to anti-PD-1/L1 immunotherapy. In a few cases, those who have shown susceptibility to immune checkpoint inhibitors do not express this marker [4,6,7]. As consequence, the selection of patients treated with immune checkpoint inhibitors is challenging in HNSCC treatment [13].

We divided the markers into four categories: checkpoint target, tumor neoantigens, tumor immune microenvironment, and radiological features.

### 3.1. Checkpoint Target

Yearley et al. studied the expression of the other ligand of PD-1, namely PD-L2. The interaction between PD-L2 and PD-1 occurs with a higher molecular affinity than that with PD-L1. Despite this, the latter is considered the main molecule responsible for PD-1-mediated immunotolerance [1,14]. PD-L2 expression usually correlates with that of PD-L1. This correlation can be traced back to one of the mechanisms of overexpression, the one that is based on genetic modifications. PD-L1 and PD-L2 are located at chromosomal locus 9p24.1, only 42 kilobases apart. It has been observed that among the mechanisms through which gene overexpression occurs, both translocation and amplification are to be counted [15]. Furthermore, the 9p24.1 locus amplification can lead to a PD-L2 overexpression through the Janus kinase 2 (JAK2)/signal transducer and activator of transcription 1 (STAT1) signaling pathway [15]. It has been observed that only PD-L2 expression is an independent predictor of response to immunotherapy in HNSCC, whereas the positivity to both PD-L1 and PD-L2 confers a greater response than PD-L1 positivity alone. PD-L2 alone was observed in 61.4–62.7% of PD-L1-negative patients and it correlates with progression-free survival (PFS) independent of PD-L1 status (Table 1). According to Wang et al., this could be one of the explanations for why some patients with PD-L1-negative malignancies respond to pembrolizumab or nivolumab therapy [16]. There is disagreement about the predictor role of PD-L2, and in some studies its values correlated with increased or decreased OS [16,17,18]. PD-L2 levels are independent predictors of progression-free survival and clinical response to pembrolizumab therapy in HNSCC patients [16,17] and when the patient is not undergoing immunotherapy, its expression is indicative of poor relapse-free survival, overall survival, and progression-free survival [17,18].

IFN-γ is considered one of the main inducers of PD-L1 expression in cancer cells. Its mechanism of action exploits the activation of STAT1, causing the expression of interferon responsive factors (IRFs) through Janus kinases (JAK-STAT pathway, especially STAT1) [38,39]. It has been observed that the inhibition of IFN-γ significantly reduces the expression of PD-L1 [40] (Table 1, Figure 1). Indeed, both PD-L1 and JAK2 reside on chromosome 9p and gene overexpression mechanisms of PD-L1 could also involve JAK2, which is, in turn, an activator of IFN-γ [1,15]. IFNs I and II activate the AKT-mTOR cascade, located downstream of the phosphatidylinositol 3 kinase (PI3K) signaling, increasing PD-L1 expression [41].

On the other hand, AKT-mTOR pathway suppression reduces IFN-γ-induced PD-L1 expression. Furthermore, the increased expression of PI3K-AKT signaling through the inhibition of its suppressor phosphatase and tensin homolog gene (PTEN) increases PD-L1 expression [41,42,43,44]. The action of IFN-γ is not limited to neoplastic cells, but also other components of the tumor mass. It has been observed that endothelial cells, under the stimulus of IFN-γ, produce PD-L1, a molecule that is otherwise not constitutively expressed [45,46]. MicroRNA post-transcriptional regulation is a crucial level of gene expression [47,48] and affects the expression of IFN-γ and PD-L1. MiR-513 binds the 3′UTR of the PD-L1 gene and inhibits its expression induced by IFN-γ, which, in turn, is an inducer of miR-513 and miR-155 [1,49,50,51]. Considering all these mechanisms, the interpretation of the level of active IFN-γ signaling as it is associated with response to PD-L1 immunotherapy is complex. A possibility is that the IFN-γ-related mRNA expression profile is a predictor of the clinical response to anti-PD-1 therapy in HNSCC [12] and it was proposed to measure the levels of IFN-γ together with those of PD-L1 to identify the subjects most likely to respond to immunotherapy [13].

Studying databases at Memorial Sloan Kettering Cancer Center and the Cancer Genome Atlas, Zhang et al. observed that predictors of response to ICIs are age and mutations of ARID1A, PIK3-CA, TP53 mutation is a negative predictor (Table 1) [34].

### 3.2. Tumor Immune Microenvironment

Hypoxia is an immune escape mechanism adopted by neoplastic cells by which an “immune desert” is generated in the tumor microenvironment. This means that the migration of T cells and their function are inhibited with their consequent lack of action. Consequently, when administering a drug inhibitor of the PD-1/L1 axis, such as pembrolizumab or nivolumab, the cells are more vulnerable to the action of T cells, but the latter are not present in the tumor microenvironment, therefore they cannot kill them. Potential markers of this phenomenon are hypoxia-inducible factor-1α (HIF-1α) and relative molecular signaling (Table 1) [21,22]. Further studies are needed to understand the correlation with response to immunotherapy in HNSCC. Like IFN-γ, EGFR is also an inducer of PD-L1, with both using JAK2 for signaling, and the overexpression of EGFR correlates with JAK2 and PD-L1 in neoplastic cells [25].

Overexpression of VEGF and TGF-β is also linked to immune tolerance. Their inhibition is associated with the recovery of the activity of the immune system against neoplastic cells observed histologically with an increase in effector T lymphocytes and a reduction in regulatory ones and myeloid-derived suppressor cells. Considering oncological therapy, the relevant data concern the newfound susceptibility to anti-PD-L1 and CTLA-4 immunotherapy by tumors in which the activity of these two molecules was suppressed [23]. The production of TGF-β has a role in both intracellular and extracellular environments. Its secretion by cancer-associated fibroblasts inhibits CD^8+^ T cells and decreases the dendritic cells in draining lymph nodes [52,53,54,55].

Overexpression of CD-73 by HNSCC cells is associated with immunosuppression of the tumor microenvironment and favors epithelial-to-mesenchymal transition and metastasis. By comparing tumors that overexpress it with those that have a lower representation, it has been seen that the presence of the molecule in high quantities is associated with reduced responsiveness to immunotherapy. Shen et al., in addition to having observed its association with the reduced response to immune checkpoint inhibitors, have also hypothesized its future role as a therapeutic target [24].

Alternative splicing analysis highlighted some expression profiles correlated with improved survival in HNSCC patients. Selected regulating splicing factors (DDX39B, PRPF39, and ARGLU1) have also been identified. Comparing the tumors that had an expression profile associated with the best prognosis with those of a control group, a different representation of the inflammatory infiltrate was observed. Therefore, the possibility of not only using this expression profile as a prognostic predictor but also as a predictor of the response to immunotherapy was hypothesized [26].

The tumor microenvironment also houses CD^8+^ T cells that are supposed to kill cancer cells. However, Chen and Mellman observed that there are tumor areas without these cells, called “immune deserts”, in which immunotherapy cannot work and, in tumors in which the inflammatory infiltrate is present, this could be non-functional [56,57]. Like PD-L1 expression, the tumor microenvironment is also subject to changes induced by therapies, and its composition should be retested to evaluate sensitivity to immunotherapy [10]. The high presence of CD^8+^ cells in the tumor microenvironment is independently associated with a lower incidence of local recurrences, and with a higher PFS and OS (Table 1) [58]. Tumor-infiltrating lymphocytes (TILs) are prognostic factors in HNSCC, their presence correlates with response to immunotherapy, and there is no agreement regarding the correlation between the expression of TILs and that of PD-L1 in HNSCC. Therefore, some authors state that TILs should be evaluated independently of PD-L1 as a prognostic factor in treatment with ICIs [59,60,61].

Tertiary lymphoid structures (TLSs) are ectopic aggregates that reproduce the structure and organization of lymphatic organs. They are also present in solid tumors with an asymmetric distribution greater in the periphery and less in the center of the mass. Some authors have hypothesized their use as predictors of response to immunotherapy [32,33].

### 3.3. Tumor Neoantigen

Usually, the immunogenicity of a neoplastic cell is also linked to the number of mutations. This, in turn, correlates with the susceptibility to the action of T lymphocytes. PD-1/L1 immunotherapy drugs simply suppress the immune escape mechanism adopted by cancer cells. In this way, their antigenicity becomes a target of lymphocytes. Indeed, it has been observed that a high tumor mutation burden (TMB-high) is a predictor of the response to anti-PD-1/L1 therapy in HNSCC [62]. There is no correlation between TMB and PD-L1, as TMB is significantly associated with PFS and treatment response in patients with immunotherapy [27]. Furthermore, the measurement of peripheral blood tumor cell mutation burden (bTMB), together with TMB and inflammatory biomarkers, is an independent predictor of pembrolizumab efficacy (Table 1) [28,29]. Progression-free survival (PFS) is significantly greater in individuals with a high number of mutations detected in circulating tumor DNA compared to those with a low number of mutations when treated with immune checkpoint inhibitors (Table 1) [30]. In a meta-analysis with over 1000 patients and 7 different cancer types, including HNSCC, Litchfield et al. analyzed predictors of response to ICIs using whole-exome and transcriptomic data. They observed that clonal TMB and TMB are the strongest predictors of response to ICIs. In their article, they considered TMB as the number of non-synonymous mutations per tumor cell and observed an odds ratio between responders to therapy, i.e., complete responders or partial responders, versus non-responders, i.e., stable disease and progressive disease, of 1.74 (95% confidence interval [1.41–2.15], *p* = 2.93). The odds ratio for total TMB was similar, with a value of 1.70 ([1.33–2.17], *p* = 1.93). Conversely, subclonal TMB was not significantly associated with TPI response [31].

The tumor microenvironment is crucial in immune tolerance [34]. The predictors of response to therapy are not only molecular but the cell ratio or the macroscopic characteristic of the mass also gives us information about the response to immunotherapy. Neutrophil-to-lymphocyte ratio (NLR) > 4 and a sum of the target lesion greater than 4 cm are significantly associated with poor response to immunotherapy and poor survival [35].

### 3.4. Radiological Features

Some radiological factors may also be useful in evaluating the immunological status of squamous cell cancer of the head and neck [10,26,32,33,56,57,58]. Starting from oral squamous cell carcinoma samples, Togo et al. observed that poor fluor-D-glucose (FDG) uptake in a PET scan is a marker of poor PD-L1 expression and CD^8+^ infiltrate in the stroma and of poor prognosis (Table 1) [36]. In particular, the most accurate metabolic parameter is the derived neutrophil-to-lymphocyte ratio [63]. Moreover, as PET visualization, magnetic resonance imaging has also been studied to identify radiological predictors of PD-L1 expression. The benchmark parameter was dynamic contrast-enhanced magnetic resonance imaging (DCE-MRI) which was correlated with PD-L1 expression. Tekiki et al. observed that the value of DCE-MRI is significantly correlated with that of PD-L1 in oral squamous cell carcinoma (Table 1) [37].

## 4. Discussion

The reviewed literature shows interesting new perspectives have emerged, which are worthy of further clinical investigation. The search for PD-L2 could be an experimental theme [17,18]. The currently approved medications for the treatment of HNSCC are molecules active against PD-1 (pembrolizumab and nivolumab) [8]. Thus, they do not account for tumor overexpression of PD-L1 or PD-L2. Although a correlation between PD-L1 and -2 was observed, this is not the rule, being homogeneous throughout the tumor tissue [17,18]. Since the prognostic value in terms of survival of PD-L2 has shown heterogeneous results, we suggest further investigation is needed [17,18]. As also noted by Wang et al., the status of PD-L2 expression by HNSCC should be considered to predict the efficacy of anti-PD-1 therapy. Furthermore, the role of PD-L2 should be investigated not only for its ability to interact with PD-1 but also because it mediates the invasion and chemoresistance of neoplastic cells [16]. The prognostic value of PD-L2 regardless of immunotherapy might be found in the actions of the molecules that are not only involved in immune escape. Among its functions is promoting the epithelial-mesenchymal transition, one of the fundamental steps for the metastasis of solid tumors [64].

One of the most promising immune checkpoint inhibitor response markers in HNSCC is IFN-γ. In fact, it has already been proposed together with PD-L1 to identify eligible patients [19,20]. IFN-γ has a high correlation with PD-L1 expression and there is widespread consensus among researchers regarding its central role in modulating PD-L1 expression. Starting from the research of Noguchi et al., it could be observed that even if the inhibition through IFN-γ antibodies considerably reduces the expression of PD-L1, this is not totally reset. Thus, we conclude that IFN-γ is critical in regulating PD-L1 expression by neoplastic cells and tumor microenvironments, but it is not the only regulatory mechanism [40].

There is a strong consensus regarding the study of gene mutations and response to immunotherapy. TMB and PD-L1, although unrelated, are both predictors of response to treatment with immune checkpoint inhibitors [27,30].

As noted by Litchfield et al., TMB appears to be the most impactful element influencing the response to ICI. This finding is not specific to HNSCC but pertains to a heterogeneous patient population affected by seven different cancer types, including head and neck cancer [31]. Therefore, there is no direct comparison between the various predictors of response to ICI, and, as stated by Burcher et al., we can observe that TMB helps us predict response to immunotherapy in HNSCC [27,28,30].

The possibility to measure TMB in its circulating form, i.e., as bTMB, highlights how liquid biopsy can be useful to investigate the characteristics of neoplasms. Circulating tumor cells have been detected in 65% of patients with HNSCC and their potential use in the diagnosis and prognosis has yet to be fully explored [65]. In the same way, infection markers such as HPV have also been identified in the plasma of patients affected by HNSCC, demonstrating how the circulating component of these neoplasms must be taken into consideration [66,67]. Circulating tumor cells in HNSCC could be used both as a prognostic factor and as a basis for analyzing the expression of molecules such as PD-L1 and other markers of response to immunotherapy [68]. Promising future results could also be achieved using non-invasive biopsy, such as salivary biomarkers [69,70].

Among other molecules identified, such as HIF-1α and JAK2, a clinical investigation is required [21,22,25]. The study of TLSs as a predictor of response to immunotherapy seems promising, but there is still no clarity about their applicability as predictors of immunotherapy. Therefore, at present, further studies are needed before incorporating this parameter into the assessment of prognosis and response to anti-PD-1/PD-L1 therapy [32,33]. TGF-β and VEGF act cooperatively in modulating the action of the immune system in the tumor microenvironment. Their ability to regulate response to immunotherapy appears to be relevant, but there are no experimental data on human patients. The studies, albeit promising, concern mouse models, so we are far from understanding their clinical applicability as predictors of the response to pembrolizumab, nivolumab, or other forms of immunotherapy currently in use. However, there are still some molecules whose value in this sense needs to be investigated with further studies [23].

CD-73 is a molecule overexpressed within tumors compared to healthy tissue. Usually, high levels correlate with a worse prognosis for patients affected by HNSCC, in terms of overall survival. This observation is partly attributed to the ability of CD-73 to promote epithelial-to-mesenchymal transition and metastasis, as well as the reduction of CD^8+^ T cell infiltration in the tumor microenvironment. Furthermore, its elevated expression is associated with a higher incidence of TP53, HRAS, and CDKN2A mutation, and negatively correlates with TMB, which is a factor of immunogenicity and a potential predictor of good response to immunotherapy [24] The analysis of alternative splicing as a prognostic factor of survival and response to immunotherapy, although interesting and compared with a control, still needs studies in larger cohorts and validation with further research [26].

The currently approved system for assessing the eligibility of HNSCC patients for the ICIs pembrolizumab and nivolumab is CPS for PD-L1 greater than or equal to 1%. This inference derives from the significant major response that the patients with this trait showed in the large trials that led to the approval of immunotherapy drugs in the treatment of HNSCC [4,6,7]. Two problems are yet to be fully understood: (i) the possibility of identifying that portion of the population below this threshold that would still respond to the therapy and (ii) how to predict which patients, while satisfying this criterion, do not respond effectively and/or durably to immunotherapy.

Some authors have observed that PD-L2 can also become a target of immunotherapy. After observing that IL-6 increases PD-L2 expression in HNSCC cell lines, they proposed to study IL-6 inhibitors in the treatment of HNSCC with high expression of this interleukin, but to date, there are no clinical trials on this subject [18]. Indoleamine 2, 3-dioxygenase (IDO) is a gene induced by IFN-γ. Some authors have hypothesized the utility of its measurement as its function in tryptophan catabolism could be one of the mechanisms by which neoplastic cells suppress T cells. Moreover, IDO increases the number of T regulatory cells (T-regs) and myeloid-derived suppressor cells (MDSCs) which further inhibit CD^8+^ T cells. There are still no conclusive studies on its correlation with any drug resistance to immune checkpoint inhibitors [10]. Some authors have already tried to create genetic scores that can serve as a guide to therapy based on prognostic-related differentially expressed ferroptosis-related genes (PR-DE-FRGs), but these still need to be validated through clinical trials [71].

The research on intratumoral PD-L1 expression by CPS has some challenges. One of these has recently been highlighted by Rasmussen et al., who observed that there is significant heterogeneity of expression within the tumor with zones of higher expression and zones of lower expression. The observed agreement within tumors for multiple biopsies using a 1% cut-off was 52% for the PD-L1 CPS [72]. In a first evaluation, some parts of the neoplastic tissue do not significantly express PD-L1, and they are therefore not susceptible to the action of immunotherapy. On the other hand, it means that a PD-L1 CPS < 1% in a single biopsy is not representative of the mass. The application of anti-PD-1 drugs to those patients, hypothetically, could cause an initial reduction of the mass which, however, would not lead to curing of the patient, but only to selecting the population of resistant neoplastic cells. This may be one explanation for the non-long-term response seen in some patients.

How PD-L1 is measured may also play a role in predicting response to therapy. Some authors have observed intra- and inter-operator variability using commercial kits (SP263 and 142) and a platform-independent test (E1L3N). The SP263 kit and the E1L3N platform were observed to have nearly perfect intra- and interobserver agreements. SP142 has a moderate interobserver agreement and reduced intra-observer agreement [73]. Furthermore, PD-L1 expression within the tumor is not constant. We know that the expression of this molecule varies and can be modified by pharmacological therapy. Radiotherapy, chemotherapy, and anti-angiogenesis agents induce the production of IFN which increases the production of PD-L1. Therefore, a tumor which at a given moment is apparently not sensitive to immunotherapy does not necessarily become so after some treatments. It, therefore, becomes useful to repeat the biopsies after having given chemotherapy and to evaluate whether the patient could be a candidate for the use of immune checkpoint inhibitors [10]. Relevant data concern the concordance between the biopsies and the surgical specimen taken from the HNSCC, in terms of the CPS for PD-L1. Up to 39% of samples analyzed by De Keukeleire et al. had discordance in this value and 34% of the samples were discordant in the measurement of TILs, further confirming that it is necessary to perform multiple and serial biopsies over time to monitor the neoplasm in the most effective way [60].

Other authors have proposed the use of the tumor proportion score (TPS) for PD-L1, as an immunotherapy response marker, but, given its correlation with the CPS, the use of which is now validated, it is necessary to verify whether its use is necessary [74]. In terms of predictors of response to therapy, we should not ignore the radiotherapy that is often used in HNSCC [75]. PD-L1 and p16 expression correlate with increased tumor radiosensitivity, while survivin and c-Met expression indicate radioresistance. These markers could be useful, together with those listed above, in evaluating the best therapy for the patient [76]. Using a machine learning mechanism that allows for a combined score across 7 tumor histotypes, a score with 11 variables has been created, highlighting that TMB and CXCL9 are the main predictors for all cancers considered [31].

Meta-analysis by Litchfield et al. analyzed 723 articles with a total pool of over 1000 patients and described several potential non-histospecific predictors of response to ICIs. In addition to TMB, they indicated several other predictors related to immunotherapy response such as frameshift insertion/deletion burden (OR = 1.38 [1.15–1.66], *p* = 1.63), nonsense-mediated decay (NMD) escaping (NMD-escape) fs-indel burden (OR = 1.38 [1.15–1.66], *p* = 5.63), proportion of mutations fitting tobacco (OR = 1.39 [1.02–1.88], *p* = 3.53), UV (OR = 1.34 [1.12–1.60], *p* = 1.23), and APOBEC (OR = 1.39 [1.09–1.76], *p* = 8.13) mutation signatures, as well as SERPINB3 mutations (OR = 1.33 [1.12–1.59], *p* = 1.23), and Fs-indel mutations escaping-NMD. Among markers of immune infiltration into the tumor microenvironment, CXCL9 expression also appears to be a predictor of response to ICIs (OR = 1.67 [1.38–2.03], *p* = 1.33). It is a chemokine that interacts with T cells by binding CXCR3, inducing the recruitment of CD^8+^ lymphocytes to the tumor microenvironment and inducing the differentiation of inflammatory T helper type 1 (Th1) and Th17 CD4 cells [31,77,78]. Additionally, CD8A (OR = 1.45 [1.20–1.74], *p* = 1.03), the T cell inflamed gene expression signature (OR = 1.43 [1.05–1.96], *p* = 2.53), and CD274 (PD-L1) expression level (OR = 1.32 [1.10–1.58], *p* = 3.03) were shown to influence response to immunotherapy [31]. We did not include these findings within the results of our study but mention them since these findings were from a heterogeneous cancer population. Although Litchfield et al. stated that most of the markers had pan-cancer significance, many molecules were not extendable to all neoplastic histotypes [31]. Furthermore, for appropriate validation, each of these predictors needs to be clinically validated in randomized trials.

Moreover, the presence of histospecific biochemical markers would make the creation of a unique score sometwhat underpowered compared to the actual possibilities that molecular medicine could offer. In fact, this would cause us to ignore or marginalize some histospecific markers which could be very useful in the therapeutic choice, such as CD38 (OR = 1.29 [1.03–1.61], *p* = 2.63), CXCL13 (OR = 1.38 [1.11–1.73], *p* = 3.83), IM-PRES (OR = 1.31 [1.05–1.65], *p* = 1.83), T effector signature from the POPLAR trial (OR = 1.38 [1.13–1.70], *p* = 1.93), and cytolytic score (OR = 1.22 [1.00–1.51], *p* = 4.93), which are not extendable to all populations of the meta-analysis. The same issue is seen for the loss of TRAF2, which increases the efficacy of ICIs by reducing the cytotoxic activity of TNF and increases T cell-induced apoptosis [31,79]. However, the loss of TRAF2 does not significantly impair the efficacy of immunotherapy in HNSCC, but it does in urothelial cancer and melanoma. Similarly, in the cohort of all the neoplasms considered it is significant, but then, specifically analyzing HNSCC, it is not [31].

The decision to adopt an immune checkpoint inhibitor therapy in R/M HNSCC may also be based on predictors other than the molecular expression of the neoplasm. Cachectic patients and those who have significant weight loss during therapy have been observed to have worse survival regardless of PD-L1 expression. Independent predictors of 6-month progression-free survival are low performance status, low subcutaneous adipose tissue level, and weight loss [80].

Analyzing the predictors, TMB and CXCL9 appear to have the greatest accuracy in predicting the response to immunotherapy in a heterogeneous population composed of patients with neoplasms of different origins [31].

Much remains to be studied on the predictors of response to immunotherapy, and it is estimated that about 40% of the factors that determine the outcome of immunotherapy have yet to be discovered [31].

## 5. Conclusions

Interesting new perspectives have emerged, regarding the already-known use of CPS for PD-L1. We suggest that, whenever possible, it is useful to look for it in multiple biopsies from the same cancer and not on a single specimen and that these should be repeated over time during treatment.

Large-scale trials are necessary to validate anti-PD-1/L1 immunotherapy markers such as PD-L2, IFN-γ, EGFR, TMB and bTMB, age, mutations of ARID1A, PIK3-CA and TP53, NLR, VEGF, TGF–β, TILs, alternative splicing, tumor size, and tumor microenvironment, including TLS, PET, and MRI contrast enhancement profiles. CD-73 seems to offer interesting perspectives both as a marker and as a potential therapeutic target, and further studies are needed. Biological predictors of radiosensitivity should also be validated to better identify patients who will respond to radiotherapy [69,70]. Studies comparing predictors appear to give greater potency to TMB and CXCR9.

## Figures and Tables

**Figure 1 diagnostics-13-00862-f001:**
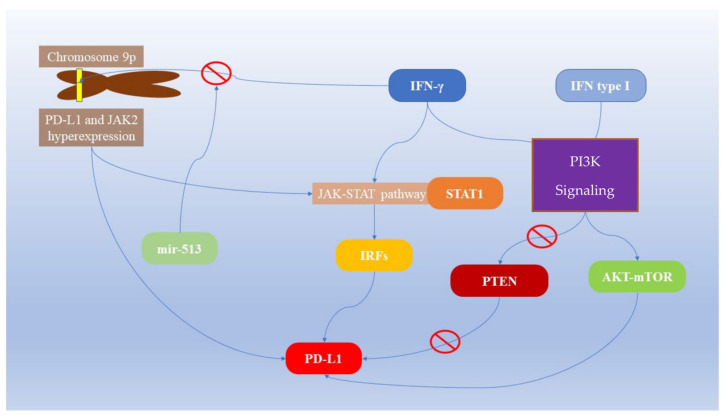
Interferon-related PD-L1 expression in cancer cells.

**Table 1 diagnostics-13-00862-t001:** Diagnostic predictors of anti-PD-1/PD-L1 immunotherapy response in HNSCC.

Predictor	Mechanism of Action/Observation	Correlation	References
PD-L2	PD-1 ligand	Correlation with PFS independent of PD-L1 status. No agreement about OS. PD-L2 levels are related to progression-free survival and clinical response to pembrolizumab therapy; whereas in patients not undergoing immunotherapy, it is related to poor relapse-free survival and progression-free survival	[16,17,18]
IFN-γ	STAT1 activation, expression of IRFs, PD-L1 induction	IFN-γ-related mRNA profile is a predictor of the clinical response to anti-PD-1 therapy in HNSCC. IFN types I and II activate PI3K which activates the AKT-mTOR cascade that is a PD-L1 inducer. It is also an inhibitor of PTEN that reduces PD-L1 levels	[19,20]
HIF-1α	Hypoxia generates an “immune desert”	HIF-1α is a marker of hypoxia *	[21,22]
VEGF and TGF-β	Immune tolerance	Increase in regulatory T cells and myeloid-derived suppressor cells	[23]
CD-73	Immunosuppression, epithelial–mesenchymal transition, metastatization	Immunosuppression in the tumor microenvironment	[24]
EGFR	PD-L1 inducer	EGFR correlates with PD-L1 expression	[25]
Alternative splicing	Immunosuppression and poor immunotherapy response	Reduction of inflammatory infiltrate in the tumor microenvironment	[26]
TMB	High TMB leads to high immunogenicity	High TMB correlates with high response to ICIs	[27,28,29,30,31]
bTMB	High bTMB relates to high TMB		
CXCL9	Interacts with T cells by binding CXCR3, inducing the recruitment of CD^8+^ lymphocytes and differentiation of inflammatory Th1 and Th17 CD4	Predictor of response to ICIs	[31]
Tertiary lymphoid structures	Potentially related to immunotherapy response	Indicators of immunogenicity	[32,33]
ARID1A, PIK3-CA	Mutations	Predictor of response	[34]
TP53	Mutation	Negative predictor of response	
NLR	NLR > 4 and major diameter of cancer > 4 cm	Associated with poor response to immunotherapy and poor survival	[35]
PET FDG uptake	Poor FDG uptake	Poor FDG uptake is a marker of poor PD-L1 expression and CD^8+^ infiltration	[36]
DCE-MRI	DCE-MRI measurement	DCE-MRI correlates with PD-L1 expression	[37]

PD-L2 = programmed cell death ligand-2; PFS = progression-free survival; OS = overall survival; PTEN = phosphatase and tensin homolog gene; PI3K = phosphatidylinositol 3 kinase; HIF-1α = hypoxia-inducible factor-1α; * = further studies necessary; TMB = tumor mutation burden; ICIs = immune checkpoint inhibitors; bTMB = peripheral blood tumor cell mutation burden; NLR = neutrophil-to-lymphocyte ratio; DCE-MRI = dynamic contrast-enhanced magnetic resonance imaging.

## Data Availability

Not applicable.

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
