# Peer review of "Diagnostic Predictors of Immunotherapy Response in Head and Neck Squamous Cell Carcinoma"

_diagnostics, 2023, doi:10.3390/diagnostics13050862_

Round 1

Reviewer 1 Report

This review analyzes the literature on Programmed Cell death ligand-1 (PD-L1) and its relationship to ICB in head and neck squamous cell carcinoma. It also discusses other potential predictors include PD-L2, IFN-γ, EGFR, VEGF, TGF–β, TMB, blood TMB, CD73, alternative splicing, tumor microenvironment, and some macroscopic and radiologic features. 

We believe this timely review lists the biomarkers of ICB and provides their potential mechanism of action. This can help the community to prioritize and focus on certain ICB biomarkers with a mechanism of action. This can be helpful to the community and especially to those who are new to the field.

  1. While the authors aim to summarize the predictors of ICB response, they have not been able to capture many most-well studied ICB predictors. Please refer to this paper for a the most comprehensive list of known biomarkers for ICB listed after surveying more than 750 studies. https://pubmed.ncbi.nlm.nih.gov/33508232/

    Exacted complete list is provided here for authors’ convenience: “The following previously published biomarkers were tested for association with response to CPI therapy: tumor mutation burden (TMB) (Rizvi et al., 2015; Snyder et al., 2014; Van Allen et al., 2015) (also split out into Clonal (McGranahan et al., 2016) and Subclonal TMB), frameshift insertion/deletion (indel) mutation burden (Turajlic et al., 2017), burden of indels escaping nonsense mediated decay (Lindeboom et al., 2019), Tobacco mutation signature (Anagnostou et al., 2020), UV signature (Knepper et al., 2019), APOBEC signature (Chapuy et al., 2019), Differential Agretopicity Index (Ghorani et al., 2018), MUC16 neoantigens (Balachandran et al., 2017), Neoantigen fitness model (Łuksza et al., 2017), SERPINB3/SERPINB4 mutations (Riaz et al., 2016), DNA damage response pathway mutations (Conway et al., 2018), Shannon diversity index for intratumor heterogeneity (SDI-ITH) (Wolf et al., 2019), burden of somatic copy number alterations (Davoli et al., 2017), burden of somatic copy number losses (Roh et al., 2017), HLA-I evolutionary divergence (Chowell et al., 2019), maximal HLA heterozygosity, HLA B44/B62 supertypes, HLA B1501 type (Chowell et al., 2018), KIR3DS1 germline variants (Trefny et al., 2019), loss of heterozygosity at the HLA locus (McGranahan et al., 2017), sex (Conforti et al., 2018), B2M mutations (Gettinger et al., 2017), JAK1/JAK2 mutations (Shin et al., 2017), KRAS and TP53 mutations (Aredo et al., 2019), PTEN mutations (Peng et al., 2016), RTK mutations (Anagnostou et al., 2020), STK11 mutations (Aredo et al., 2019), BAP1 mutations (Shrestha et al., 2019), CD8A (Tumeh et al., 2014), CD274 (PD-L1) (Gibney et al., 2016), CD38 (Chen et al., 2018), HAVCR2 (TIM3)/LGALS9 (Koyama et al., 2016), MEX3B (Huang et al., 2018) and CXCL9 expression (Chow et al., 2019), as well as the CD8 T cell effector (McDermott et al., 2018), proliferation (Pabla et al., 2019), cytolytic (Rooney et al., 2015), stroma-EMT (Wang et al., 2018), TGF beta pan fibroblast (Mariathasan et al., 2018), IMPRES (Auslander et al., 2018), CD8 T effector from the POPLAR trial (Fehrenbacher et al., 2016), 12-cheomokine (Messina et al., 2012; Tokunaga et al., 2020), HERV-3 family expression (Panda et al., 2018) and T cell inflamed gene expression signatures (Ayers et al., 2017). TMB was measured on a per megabase basis using the Friends of Cancer Research TMB Harmonization Project phase I guidelines (Merino et al., 2020), clonal TMB was measured as per (McGranahan et al., 2016) with samples which failed pyclone clustering assumed that all mutations were clonal, SCNA load was defined using the weighted genome instability index (wGII) (Endesfelder et al., 2014), expression of individual genes was measured using varianceStabilizingTransformation (vst) normalized expression count from DESeq2 (for datasets with RNaseq) or normalized nanostring expression values for the Cristescu et al. cohort. In the Cristescu et al. cohort, where transcriptome data is only available for a subset of genes, gene expression signatures were calculated with as many genes as were available. For inactivating pathway mutations (i.e., B2M, PTEN, JAK1/JAK2, DNA damage response) loss of function mutations (i.e., those causing a premature stop codon) and homozygous deletions were included. DNA damage response pathway genes were defined as: BRCA1, BRCA2, ATM, POLE, ERCC2, FANCA, MSH2, MLH1, POLD1 and MSH6 based on (Conway et al., 2018).”

  2. I think the critical weakness of this paper is that it doesn't provide the reader a way to prioritize which biomarker has a higher likelihood of success and accordingly study. This can be very helpful & resolved if the author could simply count for each biomarker the number of times an independent study has demonstrated its power during its search. Such Meta-summary can be very helpful.

    1. This can be made even stronger if the authors also gather the prediction power of the biomarker and summarize it together with a figure. We believe this can strengthen the study considerably.

  3. It would be helpful to the reader if the author systematically categorizes the biomarkers at a high level based on their type or mechanism of action. One way could be to categorize them into three classes: (i) tumor neoantigen, (ii) tumor immune microenvironment, and (iii) checkpoint target.

    1. We also recommend breaking down & structuring the review according to the categories. This will also make it easy to follow and read.

  4. While the structure, as noted in 3, can be improved, The paper is written well and clearly.

Author Response

Dear Reviewers,

We thank you for appreciating our work and for your precious suggestions that have helped us to improve it. The manuscript was edited with Microsoft Word Tracking Changes (red text) enabled to make modifications to the article easier to find.

Reviewer 1

“This review analyzes the literature on Programmed Cell death ligand-1 (PD-L1) and its relationship to ICB in head and neck squamous cell carcinoma. It also discusses other potential predictors include PD-L2, IFN-γ, EGFR, VEGF, TGF–β, TMB, blood TMB, CD73, alternative splicing, tumor microenvironment, and some macroscopic and radiologic features. 

We believe this timely review lists the biomarkers of ICB and provides their potential mechanism of action. This can help the community to prioritize and focus on certain ICB biomarkers with a mechanism of action. This can be helpful to the community and especially to those who are new to the field.”

We are happy that you appreciated the intent of our work and thank you for your precise comments. We followed your suggestions below.

Rev1.1

“While the authors aim to summarize the predictors of ICB response, they have not been able to capture many most-well studied ICB predictors. Please refer to this paper for a the most comprehensive list of known biomarkers for ICB listed after surveying more than 750 studies. https://pubmed.ncbi.nlm.nih.gov/33508232/

Exacted complete list is provided here for authors’ convenience: “The following previously published biomarkers were tested for association with response to CPI therapy: tumor mutation burden (TMB) (Rizvi et al., 2015; Snyder et al., 2014; Van Allen et al., 2015) (also split out into Clonal (McGranahan et al., 2016) and Subclonal TMB), frameshift insertion/deletion (indel) mutation burden (Turajlic et al., 2017), burden of indels escaping nonsense mediated decay (Lindeboom et al., 2019), Tobacco mutation signature (Anagnostou et al., 2020), UV signature (Knepper et al., 2019), APOBEC signature (Chapuy et al., 2019), Differential Agretopicity Index (Ghorani et al., 2018), MUC16 neoantigens (Balachandran et al., 2017), Neoantigen fitness model (Łuksza et al., 2017), SERPINB3/SERPINB4 mutations (Riaz et al., 2016), DNA damage response pathway mutations (Conway et al., 2018), Shannon diversity index for intratumor heterogeneity (SDI-ITH) (Wolf et al., 2019), burden of somatic copy number alterations (Davoli et al., 2017), burden of somatic copy number losses (Roh et al., 2017), HLA-I evolutionary divergence (Chowell et al., 2019), maximal HLA heterozygosity, HLA B44/B62 supertypes, HLA B1501 type (Chowell et al., 2018), KIR3DS1 germline variants (Trefny et al., 2019), loss of heterozygosity at the HLA locus (McGranahan et al., 2017), sex (Conforti et al., 2018), B2M mutations (Gettinger et al., 2017), JAK1/JAK2 mutations (Shin et al., 2017), KRAS and TP53 mutations (Aredo et al., 2019), PTEN mutations (Peng et al., 2016), RTK mutations (Anagnostou et al., 2020), STK11 mutations (Aredo et al., 2019), BAP1 mutations (Shrestha et al., 2019), CD8A (Tumeh et al., 2014), CD274 (PD-L1) (Gibney et al., 2016), CD38 (Chen et al., 2018), HAVCR2 (TIM3)/LGALS9 (Koyama et al., 2016), MEX3B (Huang et al., 2018) and CXCL9 expression (Chow et al., 2019), as well as the CD8 T cell effector (McDermott et al., 2018), proliferation (Pabla et al., 2019), cytolytic (Rooney et al., 2015), stroma-EMT (Wang et al., 2018), TGF beta pan fibroblast (Mariathasan et al., 2018), IMPRES (Auslander et al., 2018), CD8 T effector from the POPLAR trial (Fehrenbacher et al., 2016), 12-cheomokine (Messina et al., 2012; Tokunaga et al., 2020), HERV-3 family expression (Panda et al., 2018) and T cell inflamed gene expression signatures (Ayers et al., 2017). TMB was measured on a per megabase basis using the Friends of Cancer Research TMB Harmonization Project phase I guidelines (Merino et al., 2020), clonal TMB was measured as per (McGranahan et al., 2016) with samples which failed pyclone clustering assumed that all mutations were clonal, SCNA load was defined using the weighted genome instability index (wGII) (Endesfelder et al., 2014), expression of individual genes was measured using varianceStabilizingTransformation (vst) normalized expression count from DESeq2 (for datasets with RNaseq) or normalized nanostring expression values for the Cristescu et al. cohort. In the Cristescu et al. cohort, where transcriptome data is only available for a subset of genes, gene expression signatures were calculated with as many genes as were available. For inactivating pathway mutations (i.e., B2M, PTEN, JAK1/JAK2, DNA damage response) loss of function mutations (i.e., those causing a premature stop codon) and homozygous deletions were included. DNA damage response pathway genes were defined as: BRCA1, BRCA2, ATM, POLE, ERCC2, FANCA, MSH2, MLH1, POLD1 and MSH6 based on (Conway et al., 2018).”

Reply1.1

Thank you very much for your valuable report, the article you cited has been very useful to us, and we have not hesitated to include this evidence in the text. We further highlighted the potential role of tumor mutation burden (TMB) which was already included in our dissertation. Unfortunately, much of the evidence cited in the article by Litchfield et al. they are not, in our view, extendable to the HNSCC. They studied a pool of over 1000 patients with 7 different types of cancer. And, although they wrote that these molecules can probably be used to evaluate the susceptibility to immunotherapy in each of them, analyzing each molecule they realized that many of them were not expressed by all tumors. For this reason, we have decided to include it in the discussion and not in the results paragraph pending further unequivocal evidence regarding squamous cell carcinoma of the head and neck. The changes made to the text were done as follows:

In the results paragraph: “In a meta-analysis with over 1000 patients and 7 different cancer types, including HNSCC, Litchfield et al. analyzed predictors of response to ICIs using whole-exome and transcriptomic data. They observed that clonal TMB and TMB are the strongest predictors of response to ICI. In their article they considered TMB as the number of non-synonymous mutations per tumor cell and observed an odds ratio between responders to therapy, i.e. complete responders or partial responders, versus non-responders, i.e. stable disease and progressive disease, of 1.74 (95% confidence interval, [1.41 - 2.15], p = 2.93), the odds ratio for total TMB was similar, with a value of 1.70 [1.33–2.17], p = 1.93). Conversely, sub-clonal TMB was not significantly associated with TPI response. [31]”

In the discussion paragraph: “As noted by Litchfield et al., TMB appears to be the most impactful element influencing the response to ICI. This finding is not specific to HNSCC but pertains to a heterogeneous patient population affected by 7 different cancer types, including head and neck cancer. [31] Therefore, there is no direct comparison between the various predictors of response to ICI in the literature that is unique to HNSCC. But, as stated by Burcher et al., we can state that TMB helps us predict response to immunotherapy in HNSCC. [27,28,30]” and “Using a machine learning mechanism that allows for a combined score across 7 tumor histotypes, a score with 11 variables has been created highlighting that TMB and CXCL9 are the main predictors for all cancers considered. [31]

Litchfield et al. in their meta-analysis analyzed 723 articles with a total pool of over 1000 patients and described several potential non-histological specific predictors of response to ICI. In addition to TMB, they indicated several other predictors related to immunotherapy response such as frameshift insertion/deletion burden (OR = 1.38 [1.15 - 1.66], p = 1.63), nonsense-mediated decay (NMD) escaping (NMD-escape) fs-indel burden (OR =1.38 [1.15–1.66], p = 5.63), the proportion of mutations fitting tobacco (OR = 1.39 [1.02–1.88], p = 3.53), UV (OR = 1.34[1.12–1.60], p = 1.23), and APOBEC (OR = 1.39 [1.09–1.76], p = 8.13) mutation signatures, as well as SERPINB3 mutations (OR = 1.33 [1.12–1.59], p = 1.23), R5 fs-indelNMD-escaping mutations. Among markers of immune infiltration into the tumor microenvironment, CXCL9 expression also appears to be a predictor of response to ICI (OR = 1.67 [1.38–2.03], p = 1.33). It is a chemokine that interacts with T cells by binding CXCR3, inducing the recruitment of CD8+ lymphocytes to the tumor microenvironment and inducing the differentiation of inflammatory T helper cells type 1 (Th1) and Th17 CD4. [31,64,65]. Together with it, also CD8A (OR =1.45 [1.20–1.74], p = 1.03), the T cell inflamed gene expression signature (OR = 1.43 [1.05–1.96], p = 2.53), CD274 (PD-L1) expression level (OR = 1.32 [1.10–1.58], p = 3.03) were shown to influence response to immunotherapy. [31] We did not include these findings within the results of our study, but we still feel it is worth mentioning them in our article since these findings were made on a heterogeneous cancer population. Although Litchfield et al. stated that most of the markers had pan-cancer significance, many molecules were not extensible to all neoplastic histotypes. [31] Furthermore, for appropriate validation, each of these predictors needs to be clinically validated in randomized trials.

Moreover, the presence of histotypes-specific biochemical markers would make the creation of a unique score something underpowered compared to the actual possibilities that molecular medicine could offer. In fact, this would force us to ignore or marginalize some histo-specific markers which instead could be very useful in the therapeutic choice such as for example CD38 (OR = 1.29 [1.03–1.61, p = 2.63), CXCL13 (OR = 1.38 [1.11–1.73, p = 3.83), IM-PRES (OR = 1.31 [1.05–1.65, p = 1.83), T effector signature from the POPLAR trial (OR = 1.38 [1.13–1.70, p = 1.93), and cytolytic score (OR = 1.22 [1.00–1.51, p = 4.93), which are not extensible to all populations of the meta-analysis. The same thing also happens for the loss of TRAF2, which increases the efficacy of ICI by reducing the cytotoxic activity of TNF and increasing T cell-induced apoptosis. [31,66] However, the loss of TRAF2 does not significantly affects the efficacy of immunotherapy in HSCC, but it is in urothelial cancer and melanoma. Similarly, in the cohort of all the neoplasms considered it is significant, but then, analyzing specifically the HNSCC it is not. [31]” “Analyzing the predictors, TMB and CXCL9 appear to have the greatest accuracy in predicting the response to immunotherapy in a heterogeneous population composed of patients with neoplasms of different origins. [31]

Much remains to be studied on the predictors of response to immunotherapy, it is estimated that still, about 40% of the factors that determine the outcome of immunotherapy have yet to be discovered. [31]”

In the conclusion paragraph: “Studies comparing predictors appear to give greater potency to TMB and CXCR9.”

Many of the molecules mentioned in the second part of the comment, although great prospects for future research, have not yet been clinically tested in the HNSCC. We, therefore, excluded these markers because they are outside the scope of our paper on head and neck squamous cell carcinoma. For example, nonsense-mediated mRNA decay (NMD) was not tested on HNSCC, but on a malignancy database that included metastatic melanoma, clear cell renal cell carcinoma, and lung cancer (Lindeboom et al., 2019). Tobacco Mutation Signatures were tested in non-small cell lung cancer, UV signatures in Merkel cell carcinoma, APOBEC signatures in primary mediastinal large B-cell lymphomas (PMBLs), the Differential Agretopicity Index in lung cancer and melanoma, MUC16 neoantigens in pancreatic cancer, Neoantigen fitness model in melanoma and lung cancer, SERPINB3 and SERPINB4 in melanoma, etc. (Anagnostou et al., 2020; Knepper et al., 2019; Chapuy et al., 2019; Ghorani et al., 2018; Balachandran et al., 2017; Łuksza et al., 2017; Riaz et al., 2016)

Rev1.2

“I think the critical weakness of this paper is that it doesn’t provide the reader a way to prioritize which biomarker has a higher likelihood of success and accordingly study. This can be very helpful & resolved if the author could simply count for each biomarker the number of times an independent study has demonstrated its power during its search. Such Meta-summary can be very helpful.

    1. This can be made even stronger if the authors also gather the prediction power of the biomarker and summarize it together with a figure. We believe this can strengthen the study considerably.”

Reply1.2

We fully agree with this observation, but unfortunately, the weakness comes from the absence of studies that perform a clinical comparison of predictors. The individual publications were made in settings too heterogeneous to be compared with each other. The only article that makes this type of comparison is that of Litchfield et al. as you have previously pointed out to us in the previous comment. We included their observation that the most potent markers appear to be in TMB and CXCL9, although this was inferred on a population of 7 tumor histotypes and on HNSCC. Furthermore, unfortunately, identifying how many times a marker has been identified as a predictor in the literature is not indicative of its power, but could only be indicative of how many times it has been searched for. We inserted in our result paragraph the following sentence “In a meta-analysis with over 1000 patients and 7 different cancer types, including HNSCC, Litchfield et al. analyzed predictors of response to ICIs using whole-exome and transcriptomic data. They observed that clonal TMB and TMB are the strongest predictors of response to ICI[…]” and in the conclusion paragraph: “Studies comparing predictors appear to give greater potency to TMB and CXCR9.”

Rev1.3

“It would be helpful to the reader if the author systematically categorizes the biomarkers at a high level based on their type or mechanism of action. One way could be to categorize them into three classes: (i) tumor neoantigen, (ii) tumor immune microenvironment, and (iii) checkpoint target.

We also recommend breaking down & structuring the review according to the categories. This will also make it easy to follow and read.”

Reply1.3

Thanks for this suggestion, we have applied your indication to the paragraph of the results which is now easier to read. It has been divided into 4 parts: 3.1 Checkpoint target, 3.2 Tumor immune microenvironment, 3.3 Tumor neoantigen, and, we have inserted a fourth paragraph: 3.4 Radiological features.

Rev1.4

“While the structure, as noted in 3, can be improved, The paper is written well and clearly.”

Reply1.4

We thank you again for your advice which helped us to make the text clearer and more complete.

We hope that the changes made to the text following your instructions will satisfy you. We believe that thanks to your suggestions our manuscript has improved both in terms of structure and contents, therefore we thank you again for your precious contribution.

Reviewer 2 Report

The authors have summarized the predictors for immunotherapy response in HNSCC by searching Pubmed and Embase, etc. The author found that PD-L1 CPS is a predictor of response to immunotherapy, but it should be measured across multiple biopsies and repeatedly over time. PD-L2, IFN-γ, EGFR, VEGF, TGF–β, TMB, blood TMB, CD73, alternative splicing, tumor microenvironment, and some macroscopic and radiological features are promising predictors worthy of further studies. The manuscript is well written and here I have only several comments for the authors.

1. The authors did not compare the predicting efficacy of different predictors. Is it reasonable to compare them from different literature?

2. What about the liquid biopsy in immunotherapy response prediction?

3. It seemed that the authors did not focus on the TPS, CPS, IPS and TILs in this study. Please explain and discuss.

4. Some types of cells are reported to be predictors for therapeutic responses, such as macrophages and cancer-associated fibroblasts. Have the authors paid attention to this part?

5. The authors should provide more details about the searching procedure in database. For example, what searching terms were used in Pubmed?

Author Response

Dear Reviewer,

We thank you for appreciating our work and for your precious suggestions that have helped us to improve it. The manuscript was edited with Microsoft Word Tracking Changes (red text) enabled to make modifications to the article easier to find.

Reviewer 2

“The authors have summarized the predictors for immunotherapy response in HNSCC by searching Pubmed and Embase, etc. The author found that PD-L1 CPS is a predictor of response to immunotherapy, but it should be measured across multiple biopsies and repeatedly over time. PD-L2, IFN-γ, EGFR, VEGF, TGF–β, TMB, blood TMB, CD73, alternative splicing, tumor microenvironment, and some macroscopic and radiological features are promising predictors worthy of further studies. The manuscript is well written and here I have only several comments for the authors.”

Thank you very much for your appreciation of our manuscript. We have complied with all your comments by changing the text as below.

Rev2.1

“The authors did not compare the predicting efficacy of different predictors. Is it reasonable to compare them from different literature?”

Reply2.1

We are of the same opinion, the lack of comparisons between markers of response to immunotherapy makes it difficult to direct future research on the topic. Except for the article by Litchfield et al. which we have repeatedly mentioned in the new revision of the text, there are no studies investigating how much more effective one marker is than another. However, this paper considers a heterogeneous population of patients with 7 different malignancies. Although the authors sustain that the molecular characteristics capable of predicting the response to immunotherapy are common in these tumors, in our opinion, this statement should be taken with caution. Their own findings showed that some molecules that were significant in predicting response to immunotherapy in one tumor histotype were not significant in another. We have therefore underlined in the text that TMB and CXCL9 appear to be the most relevant, but specific comparative studies on HNSCC must be performed before reaching conclusions.

Rev2.2

“What about the liquid biopsy in immunotherapy response prediction?”

Reply2.2

We also think this is a very good point. In fact, in the text, we have mentioned the blood tumor mutation burden (bTMB) which correlates with the TMB and with the response to immunotherapy. Surely further studies will make it possible to identify numerous circulating molecules to make the liquid biopsy an always solid reality. To underline this aspect, we have added the following sentence to the bTMB discussion: “The possibility of researching TMB in its circulating form, i.e., as bTMB highlights how the liquid biopsy can be useful to investigate the characteristics of the neoplasms in the least invasive way possible. Circulating tumor cells have been detected in 65% of patients with HNSCC and their potential use in the diagnosis and prognosis of this disease has yet to be fully explored. [58] In the same way, infection markers such as HPV have also been identified in the plasma of patients affected by HNSCC, demonstrating how the circulating component of these neoplasms must be taken into consideration. [59,60] Circulating tumor cells in HNSCC could be used both as a prognostic factor and as a basis for analyzing the expression of molecules such as PD-L1 and other markers of response to immunotherapy. [61]”

Rev2.3

“It seemed that the authors did not focus on the TPS, CPS, IPS and TILs in this study. Please explain and discuss.”

Reply2.3

After briefly defining CPS and PD-L1, together with their impact on prognosis and findings from the major clinical trials involving Pembrolizumab and Nivolumab, we decided not to over-dedicate the discussion to CPS for PD-L1 as this technique is well-known and consolidated. The objective of our work is to start from the limitations of the exclusive use of CPS for PD-L1 as a marker of response to therapy and to deal with all the potential future prognostic factors that can help us make a more accurate therapeutic choice. We cited the prognostic value of tumor-infiltrating lymphocytes in our work, but after reading your comment we extended the text about it with the following sentences: “Tumor-infiltrating lymphocytes (TILs) are prognostic factors in HNSCC, their presence correlates with response to immunotherapy, and there is no agreement regarding the correlation between the expression of TILs and that of PD-L1 in HNSCC. Therefore, some authors state that TILs should be evaluated independently of PD-L1 as a prognostic factor in treatment with ICIs. [55–57].”

We also specified how TILs discordance was 34% between biopsies and operative specimens highlighting the necessity of multiple biopsies also for this analysis. “A relevant data concerns the concordance between the biopsies and the surgical material taken from the HSCC in the CPS for PD-L1. Up to 39% of samples analyzed by De Keukeleire et al. had discordance in this value and 34% of the samples were discordant in the measurement of TILs, this further confirms that it is necessary to perform multiple and serial biopsies over time to monitor the neoplasm in the most effective way. [65]”

Rev2.4

“Some types of cells are reported to be predictors for therapeutic responses, such as macrophages and cancer-associated fibroblasts. Have the authors paid attention to this part?”

Reply2.4

We paid attention to the findings on the tumor microenvironment highlighting in the results section, in paragraph 3.2, how the immune cells of the tumor microenvironment are crucial for the cytotoxic action of the immune system against the tumor. We also covered TILs more extensively thanks to your comment and described the effect of TLS on prognosis. Furthermore, in the discussion on TGF-β we have described its inhibition effect on CD8+ T lymphocytes after its secretion by cancer-associated fibroblasts with the following sentence: “The production of TGF-β has a role in both intracellular and extracellular environments. Its secretion by cancer-associated fibroblasts inhibits CD8+ T cells and decreases the dendritic cells in draining lymph nodes [52–55].”

Rev2.5

“The authors should provide more details about the searching procedure in database. For example, what searching terms were used in Pubmed?”

Reply2.5

We have added the main keywords used to search for articles in the mentioned databases to the materials and methods. “The keywords searched by the authors were: head and neck squamocellular carcinoma or head and neck squamous cell carcinoma, immunotherapy, Pembrolizumab, Nivolumab, markers of response to immunotherapy, predictors of response to immunotherapy, immune checkpoint inhibitors.”

We hope that the changes made to the text following your instructions will satisfy you. We believe that thanks to your suggestions our manuscript has improved both in terms of structure and contents, therefore we thank you again for your precious contribution.

Round 2

Reviewer 2 Report

The authors have repsonded to the comments and I do not have further suggestions.